# Chemical Components of Dufour’s and Venom Glands in *Camponotus japonicus* (Hymenoptera, Formicidae)

**DOI:** 10.3390/insects14070664

**Published:** 2023-07-24

**Authors:** Wenjing Xu, Mengqin Zhao, Lingxiao Tang, Ruoqing Ma, Hong He

**Affiliations:** 1Key Laboratory of National Forestry and Grassland Administration on Management of Forest Bio-Disaster, College of Forestry, Northwest A & F University, Xianyang 712100, China; 2Department of Entomology, South China Agricultural University, Guangzhou 510642, China

**Keywords:** ants, exocrine gland, chemical communication, GC–MS, caste differentiation

## Abstract

**Simple Summary:**

*Camponotus japonicus* is a soil-dwelling ant species that is widespread in China and Southeast Asia. A mature nest consists of minor workers, major workers, gynes, males, and a queen. The original workers, which are raised by the queen during the early nest-building stage, are smaller than other castes. The Dufour’s and venom glands are two important glands associated with the sting apparatus in female castes. Their secretions play significant roles in defense, reproduction, communication, and foraging in the social life of ants. However, the exact nature of their secretions and whether there are differences among castes remain unknown. In this study, we analyzed the secretions of these two glands using gas chromatography–mass spectrometry with two sample processing methods (hexane solution and solid-phase microextraction). The main secretion of the Dufour’s gland is n-undecane, whose proportion significantly varies among castes. The main secretions of the venom gland are formic acid and n-undecane, and their proportions show obvious differences among castes. This study provides basic information to further understand the function of these two glands in the social life of ants.

**Abstract:**

The Dufour’s and venom glands are the most developed glands connected to the female reproductive organs, playing important roles in defense, foraging, information exchange, and reproduction in ants. The main chemical secretions of these glands vary among species and even among castes of the same species. In this study, we analyzed the chemical components of the Dufour’s and venom glands in different castes of *Camponotus japonicus* (original worker, minor worker, major worker, gyne, and queen) using gas chromatography–mass spectrometry (GC–MS) with two sample processing methods (hexane solution and solid-phase microextraction). The secretion of the Dufour’s gland is characterized by a high ratio of alkanes, with n-undecane being the dominant secretion in all castes except the original workers. The venom gland’s secretion mainly includes alkanes, acids, ketones, and alcohols, with formic acid and n-undecane being the dominant components. Additionally, the chemical composition and proportion of the main components vary significantly among castes, which may be closely related to the division of labor in their social life. This study provides basic information to further understand the function of these two glands in the social life of ants.

## 1. Introduction

Ants are highly organized social insects that exhibit considerable adaptive radiation and occupy diverse ecological microhabitats [1]. They have a sophisticated communication system, particularly a chemical communication system, to ensure coordination among tens to millions of individuals [2]. This system relies on semiochemicals secreted by developed and diverse exocrine glands found throughout the ant’s body [3].

The Dufour’s and venom glands are two important glands associated with the sting apparatus in female castes. The primary function of these two glands is to store chemical secretions, but the functions of the secreted chemicals have undergone modifications in various ant taxa. The venom gland typically functions as an integral part of the venom delivery system used for paralyzing prey and defense [4]. However, the Dufour’s gland has been found to serve various functions, such as trail or territory marking, alarm–defense systems, slave-making, and sex pheromone production [5]. The chemical composition of the Dufour’s gland includes various volatile and semi-volatile compounds, such as saturated and unsaturated hydrocarbons, terpenoids, alcohols, acids, fatty acids, quinines, and aromatic compounds [6,7,8,9,10,11]. Furthermore, the chemical components of the Dufour’s gland exhibit species specificity and can be used as a chemotaxonomic clue to differentiate various ant species [12]. 

*Camponotus* is one of the largest genera in the family Formicidae, known for their larger body size and colony population compared to other species in this family. Females within the *Camponotus* species commonly exhibit behavioral polyethism based on body size or the tasks they undertake, with workers categorized as major, medium, or minor workers [13]. The chemical components of the Dufour’s gland have been investigated in over 20 *Camponotus* species to date [14,15,16,17,18,19,20,21,22,23,24,25,26,27]. However, previous research has primarily focused on workers, and there has been no comparative investigation of the chemical composition among castes.

*Camponotus japonicus* is a soil-dwelling ant species widespread in China and other areas of Asia [28]. It typically forms large colonies and forages over long distances on the ground or climbs plants to feed on honeydew from Hemipterans. A mature *C. japonicus* nest consists of minor workers (body length 9.65 ± 1.39 mm, head width 2.16 ± 0.26 mm), major workers (body length 12.6 ± 0.20 mm, head width 3.52 ± 0.03 mm), gynes (body length 14.74 ± 0.20 mm, head width 3.13 ± 0.10 mm), males (body length 9.99 ± 0.70 mm, head width 1.52 ± 0.11 mm), and queens (body length 14.74 ± 0.20 mm, head width 3.13 ± 0.10 mm) [18]. The original workers are a special caste that is reared by the queen during the early nest-building stage and are smaller than other castes [1]. The chemical composition of the Dufour’s gland in *C. japonicus* has only been studied in workers by Hayashi and Komae, using low-sensitivity detection techniques, and no differences among castes have been reported [19].

In this study, we investigate the chemical composition of the Dufour’s and venom glands in *C. japonicus* using gas chromatography–mass spectrometry (GC–MS) with two sample processing methods (hexane solution and solid-phase microextraction (SPME)). Our aim is to compare the secretion of these two glands among different female castes (minor worker, major worker, gyne, and queen), as well as the original workers. This research will provide basic information to understand the functional differentiation of these two common glands among female castes.

## 2. Material and Methods

### 2.1. Ants and Dissection

A mature nest of *C. japonicus*, including minor workers, major workers, gynes, and queens, was excavated in Yangling, Shaanxi province, located at approximately 34°18′13.88″ N latitude and 108°04′34.72″ E longitude. In addition, we collected 10 new queens from a nuptial flight site in Yangling and reared them until the original workers emerged. Randomly selected individuals from each caste (minor workers, major workers, gynes, queens, and original workers) were analyzed for the chemical composition of their Dufour’s and venom glands.

The production of formic acid is one of the primary functions of the venom gland in Formicinae ants. However, formic acid is highly volatile, and its chromatographic peak would overlap with the solvent peak in the hexane solution method. Therefore, we combined the solid-phase microextraction (SPME) method to analyze the secretion of the Dufour’s and venom glands and used a standard substance to quantify formic acid.

The dissection of the Dufour’s and venom glands followed the methods described by Chen et al. [29]. Individual ants were first placed in a −20 °C refrigerator for approximately 10 min to reduce their activity and then dissected under a stereomicroscope (Leica EZ4D, Germany) in a 0.65% NaCl solution. Each Dufour’s or venom gland from an individual ant was treated as one sample. For the SPME method, the samples were stored in a soft glass capillary (diameter 2 mm × wall thickness 0.2 mm × length 100 mm), with 7 duplications for minor workers and original workers, 8 for major workers and gynes, and 10 for queens. An empty capillary was used as a blank. Before detection, the prepared samples were first thawed for 30 min at room temperature, and the capillaries were carefully cut at the distal end. The conditioned SPME fibers (65 μm PDMS/DVB, stableflex 24Ga, Manual Holder, 3pk) were then inserted and allowed to absorb the headspace volatiles for 30 min. For the hexane solution method, the glands were placed into vials with 50 μL of hexane (≥99.5% purity) and extracted for 2 h at room temperature. Afterward, the gland tissues were removed with a clean needle, and 300 ng of n-tetradecane (which was proved not to exist in the gland secretion profiles through trial testing) was added as an internal reference. Three replicates were prepared for each caste, and vials with only 50 μL of hexane and 300 ng of n-tetradecane served as blanks. The blanks and all samples were stored together at −80 °C until detection.

### 2.2. Gas Chromatography–Mass Spectrometry (GC–MS)

The samples were analyzed using a GC–MS system (TRACE 1310 and ISQ Single Quadrupole MS, Thermo Fisher Scientific, Waltham, MA, USA). The instrument was equipped with a low-polar HP-5MS UI capillary column (30 m × 0.25 mm, 0.25 µm film thickness). The mass spectra were operated in EI mode at 70 eV, with an ion source and transmission line temperature of 280 °C. The chromatographic conditions included a sample pool temperature of 180 °C, and the column was programmed from 40 °C (held for 2 min) to 300 °C at a rate of 8 °C/min, with a 10 min hold at 300 °C. The hexane solvent samples, standards of C7–C40 alkanes, and formic acid were analyzed using the same instruments and procedure.

The compounds were identified using a combination of co-injection with standards (for formic acid and C7–C39 alkanes), retention index (calculated using the temperature programming formula: I = 100Z + 100[TR(x) − TR(z)]/[TR(z + 1) − TR(z)], where TR represents retention temperature, x represents the carbon number of the target compound, and z and z + 1 represent the carbon number of the n-alkanes before or after the target compound), and mass spectra comparisons to the NIST05 databases. The quantity of compounds detected with the hexane solution method was calculated using the peak area ratio with the internal reference of n-tetradecane. The relative percentage was calculated based on the peak area of each compound relative to the total sample area after removing impurities detected in the blanks.

An UpSet plot was constructed in R using the UpSet package to illustrate the similarities and differences in the composition of gland secretions among castes [30]. The peak areas were subjected to a variance analysis with ANOVA followed by Duncan’s multiple range test (*p*-value < 0.05) using JMP Pro 16. The differences were visualized using PCA analysis in R (ggplot2). Summarized chromatographic plots were generated using Origin 2021, displaying the average peak areas.

## 3. Results

### 3.1. Chemical Components of Dufour’s Gland Secretions

Our study utilized the SPME and hexane solution methods to collect and analyze the secretions from the Dufour’s gland in *C. japonicus*. Through GC–MS analysis, we identified a complex blend of 31 compounds in the gland secretions, including chemical signatures of all five castes (minor worker, major worker, original worker, gyne, and queen) (Table 1 and Table 2). Among these compounds, 21 were shared among the castes. The Dufour’s gland secretion of minor workers exhibited the highest component richness with 30 compound species, while the queen had the lowest with 23 compound species (Figure 1). The SPME method detected 6 compounds (Table 1, Appendix A), while the hexane solution method detected 30 compounds (Table 2, Appendix A). Four alkane compounds (n-decane, n-undecane, n-dodecane, and n-pentadecane) and 8-heptadecene can be detected in both methods.

In the SPME method results, n-undecane was the main secretion in all four detected castes (minor worker, major worker, gyne, and queen), with relative percentages ranging from 86% to 90%, and no significant difference was observed among them. N-pentadecane and 8-heptadecene were particularly enriched in the queens. N-tridecane was identified as a major secretion in the Dufour’s gland of the gynes and was significantly higher than in other castes (Table 1, Appendix A).

Using the hexane solution method, a total of 30 compounds were detected in the Dufour’s gland of the original worker, minor worker, major worker, gyne, and queen of *C. japonicus*, with respective numbers of 27, 30, 25, 28, and 22 compounds. The weight of the substances detected in the Dufour’s gland of gynes was the highest, while the lowest was in the queens (7108.63 ng and 1730.94 ng, respectively) (Table 2). Similarly, to the SPME method, alkanes were the most abundant compounds in all analyzed castes, with n-undecane remaining the dominant compound. However, the content of n-undecane was significantly higher in gynes compared to other castes (Table 2). Additionally, lower levels of alcohols, esters, and acids were also detected, with variations in content among the castes. The original worker caste exhibited the most distinct composition, with 11 significantly enriched components, especially squalene (Table 2, Appendix A). In contrast, the queen caste only had 22 compounds detected, with their contents being generally lower than in other castes. Overall, the Dufour’s gland secretion could discriminate among the different castes (Figure 2A).

### 3.2. Chemical Composition of Venom Gland Secretions

In the venom gland of *C. japonicus*, a total of 24 compounds have been identified using two sample processing techniques (Table 3 and Table 4). Only 9 compounds were detected using the SPME method (Table 3, Appendix A), while 19 compounds were detected using the hexane solution method (Table 4, Appendix A). Among these compounds, four substances (n-decane, n-undecane, n-pentadecane, and 1-hexadecanol) were detected using both methods. Most of the compounds were shared among the different castes (Figure 3).

In the results of the SPME method, formic acid was the dominant secretion, accounting for more than 90% of the identified compounds, and it showed significant differences among all five castes. The relative percentage of formic acid was the highest in minor workers, with 99.11%, and the lowest in queens, with 90.96% (Table 3). Apart from formic acid, a certain amount of n-undecane and 3-penten-2-one were also detected, showing no significant difference among the castes (Table 3).

Regarding the results of the hexane solution method, 19 compounds were detected from the venom gland of *C. japonicus*, but formic acid could not be detected using this method. When the internal reference n-tetradecane was eliminated, the weight of all other substances was very low. The total weight of secretions in the venom gland was the highest in gynes (336.09 ng) and the lowest in minor workers (35.02 ng) (Table 4). Some substances, such as n-undecane, were significantly higher in gynes, squalene was remarkably enriched in queens and original workers, and the content of 9E-hexadecen-1-ol was particularly high in queens (Table 4). However, the composition of these compounds accounted for a very small proportion of the overall secretion profile of the venom gland, making it difficult to distinguish castes based on the venom gland secretion profile in *C. japonicus* (Figure 2B).

## 4. Discussion

The Dufour’s gland and venom gland are two important glands in Formicidae. The chemical compounds of these glands often vary among species and even within different castes of the same species. In our study, we investigated the chemical composition of the Dufour’s and venom glands in *C. japonicus* and compared their variations among four female castes (minor worker, major worker, gyne, and queen) and the original workers. The results showed that both glands are composed of complex chemical components, and the content of some main compounds slightly or obviously varied among castes.

### 4.1. The Components and Function of Dufour’s Gland in Camponotus Ants

Alkanes are the main components of the Dufour’s gland in most *Camponotus* species, with n-undecane and n-tridecane usually being the major compounds (Appendix A). N-undecane is typically an active alarm pheromone in most *Camponotus* species, except for cases such as *C. herculeanus*, *C. ligniperda*, *C. intrepidus*, *C. sericeus*, *C. lateralis rebeccae*, and *C. lateralis*—where n-tridecane is the dominant secretion [15,16,17,20,23]—and the cases of *C. gestroi* and *C. balzani*—where the main secretions are 5-methyl-tridecane and octyl hexanoate [20,26]. Our results confirm that the main secretion of *C. japonicus* is consistent with most species in this genus, with n-undecane being the dominant compound in all analyzed castes [12].

The secretion of the Dufour’s gland in *C. japonicus* workers was previously studied by Hayashi and Komae, who found that almost 100% of the content was alkanes, with n-undecane being the major component, accounting for over 97% in workers [19]. In our study, in addition to workers, we also analyzed the secretion of the Dufour’s gland in queens, gynes, and the original workers. We detected additional components besides n-undecane, which remains the dominant substance. As a result, the relative content of n-undecane is lower than the results reported by Hayashi and Komae to varying degrees [18]. N-undecane is a well-known component that usually functions as an alarm pheromone in ants [1,31]. It also acts as a signal to attract males during nuptial flights, as observed in *Formica lugubris* [32]. In *C. japonicus*, n-undecane is most abundant in gynes and least abundant in queens. This leads us to speculate that n-undecane may act as a sex-attracting pheromone in gynes and as an alarm pheromone in workers.

### 4.2. The Components and Function of the Venom Gland in C. japonicus

The venom gland, also known as the poison gland, primarily serves roles in predation and defense [1]. Formic acid, the simplest organic acid, is the typical secretion of the venom gland in ants of the subfamily Formicinae. Previous studies have shown that the venom gland of *Camponotus* species usually contains only formic acid [20,27]. However, in the venom gland secretion of *C. japonicus*, we found high levels of n-undecane and other substances such as acids, alcohols, alkanes, alkenes, benzene, esters, and phenols.

Results of solid-phase microextraction (SPME) showed that the main secretion of the venom gland in *C. japonicus* consists of formic acid and n-undecane, with a remarkable difference in the content of formic acid among castes. N-undecane is the dominant substance in both the venom gland and Dufour’s gland. These two compounds are widely used in the ants’ alarm–defense system [1,33]. Fujiwara-Tsujii et al. (2006) studied the specific functions of these two compounds in *C. obscuripes* [27]. The results demonstrated that the function of formic acid is an alarm pheromone, causing workers to flee from the source after receiving the signal. On the contrary, n-undecane acts as an aggregation pheromone, causing workers to move towards the source after receiving the signal. We believe that the secretions of the venom gland in worker castes of *C. japonicus* may exhibit similar functions to those observed in *C. obscuripes*. Additionally, formic acid can regulate the pH in the digestive system, aiding in gut microbial selection or creating a favorable microhabitat for the ants, as observed in the queens of *C. japonicus* [34]. We wish that more behavioral studies will be conducted to confirm the function of formic acid and n-undecane in *C. japonicus*.

### 4.3. The Volatile Secretions and Functional Adaptability of Dufour’s and Venom Gland in the Original Worker

In our study, we also analyzed the chemical components of the Dufour’s and venom glands in the original worker, which is a special developmental stage caste that is raised by the new queen herself. They grow and develop using the metabolites stored in the alary muscles and fat bodies of the mother colony. During the stage when a new queen establishes a new nest, she must allocate the limited energy stored in her body wisely to ensure both her own survival and the production of offspring [1]. Therefore, all the structures in original workers serve the most basic functions for survival. The secretion composition of the Dufour’s gland in the original workers is the most distinctive among all castes. The small body size of the original worker determines its alarm strategy. Considering their disadvantage in fighting, they prefer to call more nestmates to deal with dangers collectively. Thus, in addition to n-undecane, a significant amount of another aggregation pheromone, squalene, is secreted by both the Dufour’s gland and the venom gland in the original worker. Squalene is found in the cuticular hydrocarbons of coffee beans and is known to be involved in nestmate aggregation and communication [35,36]. The presence of squalene in the glands of the original worker suggests its potential role in enhancing nestmate recognition and collective defense behavior.

In conclusion, the chemical composition of the Dufour’s and venom glands in *C. japonicus* exhibited variations among different castes, indicating their functional adaptability. N-undecane is the dominant compound in both glands and is likely involved in alarm, aggregation, and sex-attracting pheromone signaling. Formic acid is another important component of the venom gland and has multiple functions, including defense and maintaining digestive system pH. The original worker, with its unique secretion composition, likely utilizes squalene for nestmate recognition and collective defense. Further studies are needed to explore the precise functions and mechanisms of these chemical components in different castes of *C. japonicus* and their contribution to the overall social behavior of the species.

## Figures and Tables

**Figure 1 insects-14-00664-f001:**
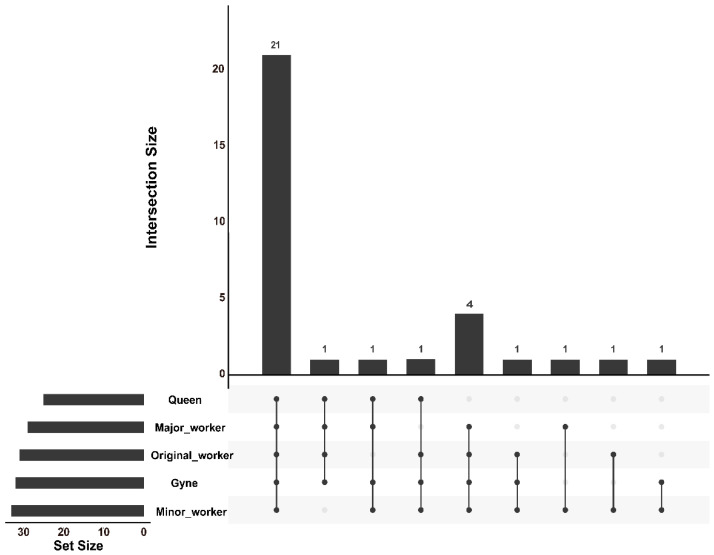
UpSet plots showing Dufour’s gland secretion compounds overlap among different castes in *C. japonicus*. The black dots indicate inclusion of this caste, and the gray dots indicate exclusion of this caste. The bars mean the number of compounds. The data were from results of SPME and hexane solution methods.

**Figure 2 insects-14-00664-f002:**
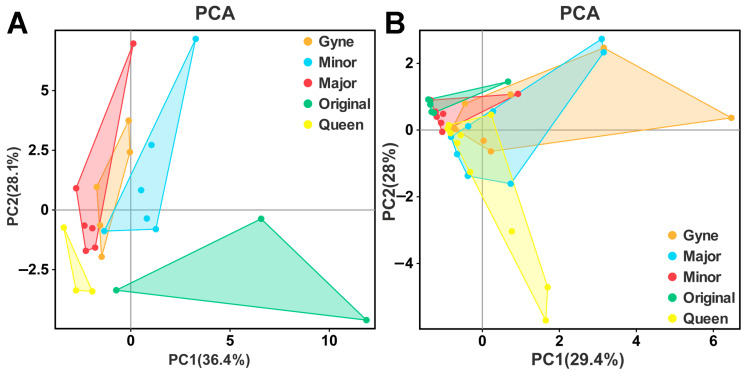
PCA plots based on peak areas. (**A**) is for Dufour’s gland based on hexane solution method data. (**B**) is for venom gland based on SPME method data. Major: major worker; Minor: minor worker; Original: original worker.

**Figure 3 insects-14-00664-f003:**
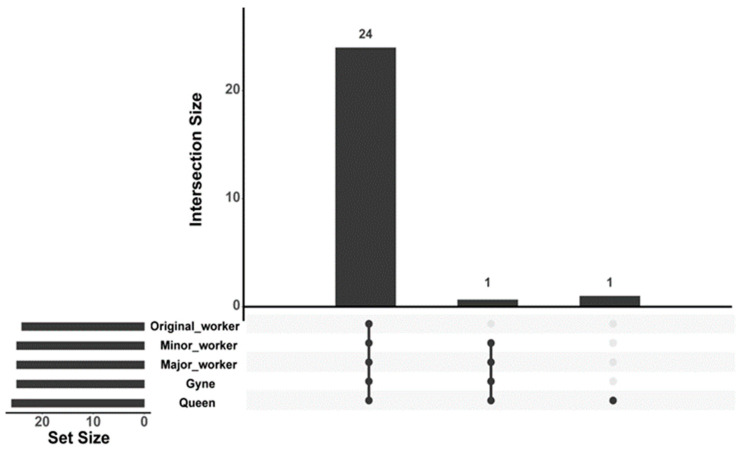
UpSet plots to show venom gland secretion compounds overlap among different castes in *C. japonicus*. The black dots indicate inclusion of this caste, and the gray dots indicate exclusion of this caste. The bars mean the number of compounds. The data were from results of SPME and hexane solution methods.

**Table 1 insects-14-00664-t001:** Chemical components of Dufour’s gland secretions of *C. japonicus* detected with SPME method.

Compound	CAS Number	Molecular Formula	*p*-Value	Retention Index	Identification Standard	CN	Relative Percentage (%) of Each Compound			
							Minor Worker	Major Worker	Gyne	Queen
Alkane										
**n-Decane**	124-18-5	**C_10_H_22_**	**0.3276**	**1000**	**KI & S & ms**	**1**	**6.38 a**	**6.17 a**	**2.85 a**	**4.91 a**
**n-Undecane**	**1120-21-4**	**C_11_H_24_**	**0.4891**	**1100**	**KI & S & ms**	**2**	**84.17 a**	**87.82 a**	**86.78 a**	**77.41 a**
**n-Dodecane**	**112-40-3**	**C_12_H_26_**	**0.0337**	**1200**	**KI & S & ms**	**3**	**0.66 ab**	**0.53 a**	**0.59 a**	**0.36 b**
n-Tridecane	629-50-5	C_13_H_28_	0.1878	1300	KI & S & ms	4	6.81 ab	4.65 ab	8.39 a	3.76 b
**n-Pentadecane**	**629-62-9**	**C_15_H_32_**	**<0.0001**	**1500**	**KI & S & ms**	**5**	**1.76 b**	**0.77 b**	**1.29 b**	**13.22a**
Alkene										
**8-Heptadecene**	**2579-04-6**	**C_17_H_34_**	**0.0605**	**1681**	**KI & ms**	**6**	**0.23 ab**	**0.07 b**	**0.1 ab**	**0.34 a**

CN: compound number in Appendix A; compounds detected using both methods are written in bold. The bold line means overlap with Table 2. The Dufour’s gland of original worker is too small to detect any substance using SPME method. Data in the same column marked with different letters indicate significant differences (Duncan’s multiple range test, 0.05 level).

**Table 2 insects-14-00664-t002:** Chemical components of Dufour’s gland secretions in *Camponotus japonicus* detected using hexane solution method.

	CAS Number	Molecular Formula	*p*-Value	RI	IS	CN	Quantity (ng)/Relative Percentage (%) of Each Compound				
							Original Worker	Minor Worker	Major Worker	Gyne	Queen
Acids											
(Z)-13-Octadecenoic acid	13126-39-1	C_18_H_34_O_2_	0.1324	3058	ms		16.38/0.35 a	11.46/0.19 ab	8.64/0.18 b	6.34/0.09 b	5.97/0.34 b
11-Octadecenoic acid	506-17-2	C_18_H_34_O_2_	0.2619	3152	ms	12	93.7/2.01 a	181.55/3.03 a	169.9/3.45 a	91.4/1.28 a	9.31/0.53 a
(Z)-9-Hexadecenoic acid	373-49-9	C_16_H_30_O_2_	0.3160	2963	ms		13.25/0.28 a	9.22/0.15 a	7.12/0.14 a	4.71/0.07 a	4.29/0.25 a
Alcohol											
n-Heptadecan-1-ol	1454-85-9	C_17_H_36_O	0.0333	1892	ms	6	76.03/1.63 a	4.55/0.08 b	3.58/0.07 b	0.9/0.01 b	0/b
n-Nonadecan-1-ol	1454-84-8	C_19_H_40_O	0.0288	2156	KI & ms	8	249.27/5.35 a	6.14/0.10 b	0/b	0/b	0/b
(E)-Geranylgeraniol	24034-73-9	C_20_H_34_O	0.1628	2488	ms	9	21.33/0.46 ab	17.03/0.28 ab	0/b	16.22/0.23 a	0.89/0.05 b
Hexadecan-1-ol	36653-82-4	C_16_H_34_O	0.0242	1891	ms		4.7/0.10 b	3.19/0.05 b	3.4/0.07 b	8.47/0.12 a	0/b
2-octyl-Dodecan-1-ol	5333-42-6	C_20_H_42_O	0.0128	3029	ms		14.44/0.31 a	7.97/0.13 b	5.97/0.12 b	5.4/0.08 b	5.26/0.30 a
(Z)-8-Dodecen-1-ol	40642-40-8	C_12_H_24_O	<0.001	1674	ms		0/b	1.05/0.02 b	4.13/0.08 ab	0/a	0/b
Alkane											
**n-Decane**	**124-18-5**	**C_10_H_22_**	**0.0226**	**1000**	**KI & S & ms**	**1**	**19.14/0.41 ab**	**73.52/1.23 a**	**47.71/0.97 ab**	**67.52/0.95 a**	**6.09/0.35 b**
**n-Dodecane**	**112-40-3**	**C_12_H_26_**	**0.1802**	**1200**	**KI & S & ms**	**3**	**98.63/2.12 b**	**339.81/5.67 a**	**238.58/4.84 ab**	**218.78/3.07 ab**	**113.19/6.47 ab**
n-Heptadecane	629-78-7	C_17_H_36_	0.1665	1700	KI & S & ms	5	30.32/0.65 a	78.56/1.31 a	68.14/1.38 a	32.34/0.45 a	27.84/1.59 a
n-Nonadecane	629-92-5	C_19_H_40_	0.0903	1900	KI & S & ms		7.9/0.17 ab	5.15/0.09 a	4/0.08 ab	1.08/0.02 ab	0/b
n-Octadecane	593-45-3	C_18_H_38_	0.0024	1800	KI & S & ms		11.98/0.26 a	4.13/0.07 b	0/b	0.96/0.01 b	1.84/0.11 b
n-Tritriacontane	630-05-7	C_33_H_68_	0.6228	3300	KI & S & ms		15.83/0.34 a	13.96/0.23 a	10.64/0.22 a	10.42/0.15 a	10.22/0.58 a
n-Pentadecane	629-62-9	C_15_H_32_	0.0282	1500	KI & S & ms	4	54.55/1.17 b	220.14/3.67 a	179.76/3.65 ab	67.12/0.94 b	43.83/2.51 b
n-Tetratriacontane	14167-59-0	C_34_H_70_	<0.001	3400	KI & S & ms		17.39/0.37 a	14.23/0.24 ab	11.4/0.23 b	10.63/0.15 b	10.51/0.60 b
**n-Undecane**	**1120-21-4**	**C_11_H_24_**	**0.0747**	**1100**	**KI & S & ms**	**2**	**1982.52/42.59 b**	**4613.94/77.01 ab**	**4007.82/81.35ab**	**6457.91/90.54 a**	**1394.24/79.70 b**
2-methyl-Octadecane	1560-88-9	C_19_H_40_	0.0335	2596	ms		8.58/0.18 ab	5.37/0.09 a	3.08/0.06 bc	2.79/0.04 bc	2.57/0.15 c
3-methyl-Undecane	1002-43-3	C_12_H_26_	0.0235	1171	ms		0/a	3.15/0.05 a	3.17/0.06 a	3.71/0.05 a	5.33/0.30 a
7-hexyl-Eicosane	55333-99-8	C_26_H_54_	0.1315	3126	ms		15.2/0.33 a	10.99/0.18 b	8.37/0.17 c	7.74/0.11 c	7.56/0.43 c
8-hexyl-Pentadecane	13475-75-7	C_21_H_44_	0.0477	3437	ms		18.46/0.40 a	13.81/0.23 b	11.05/0.22 b	10.15/0.14 b	9.64/0.55 b
Alkene											
Squalene	111-02-4	C_30_H_50_	<0.001	2921	S & ms	11	1427.94/30.67 a	228.08/3.81 b	24.17/0.49 b	39.28/0.55 b	41.61/2.38 b
1-Undecene	821-95-4	C_11_H_22_	0.1730	1093	ms		0/b	2.2/0.04 a	0/b	3.21/0.04 a	0/b
8-Heptadecene	2579-04-6	C_17_H_34_	0.0311	1681	KI & ms		5.03/0.11 a	35.23/0.59 a	54.1/1.10 a	14.02/0.20 a	12.65/0.72 a
9-Nonadecene	31035-07-1	C_19_H_38_	0.3622	1877	ms		4.82/0.10 a	3.06/0.05 a	2.05/0.04 a	0.95/0.01 a	0/b
Ester											
Octadecyl acetate	822-23-1	C_20_H_40_O_2_	0.0365	2315	ms		5.33/0.11 a	3.54/0.06 ab	0/b	5.33/0.07 a	0/b
Hexadecyl dodecanoate	20834-06-4	C_28_H_56_O_2_	0.0695	3075	ms		14.36/0.31 a	11.2/0.19 a	7.46/0.15 a	8.43/0.12 a	5.66/0.32 a
Hexadecyl 2-ethylhexanoate	59130-69-7	C_24_H_48_O_2_	0.0022	2539	ms		21.41/0.46 a	5.41/0.09 b	2.83/0.06 b	4.55/0.06 b	3.14/0.18 b
Isopropyl hexadecanoate	142-91-6	C_19_H_38_O_2_	<0.001	2126	ms	7	323.83/6.96 a	24.66/0.41 b	11.86/0.24 b	8.27/0.12 b	9.3/0.53 b
Quantities in total (ng)							4572.32	5952.3	4898.93	7108.63	1730.94

CN: compound number in Appendix A; IS: identification standard; RI: retention indices; compounds detected using both methods are written in bold. The bold line means overlap component with Table 1. *p*-value < 0.05. Data in the same column marked with different letters indicate significant differences at the 0.05 level. (Duncan’s multiple range test).

**Table 3 insects-14-00664-t003:** Chemical components of venom gland secretions of *Camponotus japonicas* detected using SPME method.

Compound	CAS Number	Molecular Formula	*p*-Value	RI	IS	CN	Relative Percentage (%) of Each Compound				
							Original Worker	Minor Worker	Major Worker	Gyne	Queen
Acid											
Formic acid	64-18-6	CH_2_O_2_	<0.0001		S & ms	1	99.06 d	99.11 c	95.07 ab	99.06 b	90.96 a
Hexanoic acid	142-62-1	C_6_H_12_O_2_	0.1485	981	ms	4	0.07 a	0.01 a	0.01 a	0.01 a	0.03 a
2,5-dihydroxy-Benzeneacetic acid	451-13-8	C_8_H_8_O_4_	0.3815	1624	ms	9	- a	- a	- a	- a	0.04 a
Alcohol											
**1-Hexadecanol**	**36653-82-4**	**C_16_H_34_O**	**0.1548**	**1891**	**ms**	**10**	**- a**	**- a**	**0.01 a**	**0.01 a**	**0.01 a**
Alkane											
**n-Decane**	124-18-5	**C_10_H_22_**	**0.2387**	**1000**	**KI & S & ms**	**5**	**- a**	**- a**	**- a**	**- a**	**0.01 a**
**n-Pentadecane**	629-62-9	**C_15_H_32_**	**0.0088**	**1500**	**KI & S & ms**	**8**	**- b**	**0.01 b**	**0.12 a**	**0.05 a**	**- b**
n-Tridecane	629-50-5	C_13_H_28_	0.0929	1300	KI & S & ms	7	- b	0.02 b	0.51 a	0.04 ab	0.01 b
**n-Undecane**	**1120-21-4**	**C_11_H_24_**	**0.3238**	**1100**	**KI & S & ms**	**6**	**0.41 a**	**0.72 a**	**4.11 a**	**0.78 a**	**0.62 a**
Ketone											
3-Penten-2-one	625-33-2	C_5_H_8_O	<0.0001		ms	3	0.45 d	0.12 cd	0.18 bc	0.06 a	8.33 ab

CN: compound number in Appendix A; IS: identification standard; RI: retention indices; compounds detected using both methods are written in bold. The bold line means overlap component with Table 4. *p*-value < 0.05. Data in the same column marked with different letters indicate significant differences at the 0.05 level. (Duncan’s multiple range test).

**Table 4 insects-14-00664-t004:** Chemical components of venom gland secretions of *Camponotus japonicus* detected using hexane solution method.

	CAS Number	Molecular Formula	*p*-Value	RI	IS	CN	Quantity (ng)/Relative Percentage (%) of Each Compound				
							Original Worker	Minor Worker	Major Worker	Gyne	Queen
Acids											
Pterin-6-carboxylic acid	948-60-7	C_7_H_5_N_5_O_3_	0.1427	1335	ms	5	2.83/2.84 a	1.01/2.72 a	5.44/3.20 a	7.03/2.02 a	3.34/2.20 a
(Z)-11-Octadecenoic acid	506-17-2	C_18_H_34_O_2_	0.3271	3152	ms	17	3.93/3.95 c	4.10/11.01 a	13.29/7.82 b	10.06/2.89 c	8.60/5.66 bc
Alcohol											
9E-Hexadecen-1-ol	64437-47-4	C_16_H_32_O	0.1266	1958	ms	10	3.17/3.19 b	1.07/2.87 b	5.23/3.08 b	26.31/7.56 ab	28.33/18.64 a
Hexadecan-1-ol	**36653-82-4**	**C_16_H_34_O**	**0.0681**	**1891**	**ms**	**9**	**4.00/4.02 ab**	**1.05/2.81 b**	**5.13/3.02 b**	**5.22/1.5 a**	**3.13/2.06 ab**
Alkane											
**n-Decane**	**124-18-5**	**C_10_H_22_**	**0.0174**	**1000**	**KI & S & ms**	**2**	**3.20/3.22 c**	**1.31/3.52 bc**	**15.67/9.22 a**	**13.92/4.00 ab**	**3.89/2.56 c**
n-Hentriacontane	630-04-6	C_31_H_64_	0.0423	3220	KI & S & ms	18	5.06/5.08 c	1.40/3.77 bc	14.92/8.78 abc	12.11/3.48 a	9.88/6.50 ab
**n-Pentadecane**	**629-62-9**	**C_15_H_32_**	**0.6784**	**1500**	**KI & S & ms**	**8**	**2.74/2.75 a**	**1.34/3.59 a**	**5.17/3.04 a**	**6.61/1.90 a**	**3.09/2.03 a**
n-Tritriacontane	630-05-7	C_33_H_68_	0.0862	3413	KI & S & ms	19	5.59/5.62 b	1.47/3.95 ab	5.73/3.37 ab	4.38/1.26 a	3.98/2.62 ab
**n-Undecane**	**1120-21-4**	**C_11_H_24_**	**0.0065**	**1100**	**KI & S & ms**	**3**	**7.35/7.39 b**	**9.48/25.46 b**	**13.80/8.12 b**	**145.71/41.87 a**	**5.78/3.80 b**
2-methyl-Eicosane	1560-84-5	C_21_H_44_	0.0600	3001	ms	16	2.61/2.62 b	2.55/6.86 b	12.17/7.16 ab	21.65/6.22 a	7.33/4.82 ab
n-Octadecane	630-02-4	C_28_H_58_	0.0489	2800	KI & S & ms	14	5.31/5.34 c	1.53/4.12 bc	10.28/6.05 ab	13.64/3.92 a	4.24/2.79 abc
2,2,4,6,6-pentamethyl-Heptane	13475-82-6	C_12_H_26_	0.0108	988	ms	1	6.06/6.09 a	0.93/2.49 b	10.72/6.31 b	12.11/3.48 b	6.11/4.02 a
n-Tridecane	629-50-5	C_13_H_28_	0.0093	1300	KI & S & ms	4	2.85/2.86 bc	1.28/3.44 ab	9.16/5.39 c	7.24/2.08 a	5.38/3.54 ab
8-hexyl-Pentadecane	13475-75-7	C_21_H_44_	0.0463	3436	ms	20	5.25/5.28 c	1.44/3.88 bc	5.85/3.44 abc	3.97/1.14 ab	4.00/2.63 a
Alkene											
Squalene	111-02-4	C_30_H_50_	0.0030	2921	S & ms	15	21.65/21.75 a	2.07/5.56 b	9.40/5.53 b	8.28/2.38 b	37.48/24.66 a
Ester											
Hexadecyl acetate	629-70-9	C_18_H_36_O_2_	0.0111	2114	ms	11	3.16/3.18 b	1.06/2.84 b	5.83/3.43 b	25.54/7.34 a	3.06/2.01 b
Ketone											
bis-1,1′-(1,4-phenylene) Ethanone	1009-61-6	C_10_H_10_O_2_	0.2318	1482	ms	7	2.70/2.71 a	0.98/2.62 a	5.34/3.14 a	6.54/1.88 a	2.99/1.97 a
Phenol											
2-methyl-1,4-Benzenediol	95-71-6	C_7_H_8_O_2_	0.3459	1371	ms	6	2.49/2.50 a	0.95/2.56 a	5.01/2.95 a	6.16/1.77 a	3.18/2.09 a
Others											
(Z)-9-Octadecenamide	301-02-0	C_18_H_35_NO	0.0051	2487	ms	13	4.06/4.08 b	1.13/3.04 b	6.39/3.76 b	6.26/1.80 b	5.24/3.45 a
Quantities in total (ng)							89.50	35.02	158.09	336.09	143.32

CN: compound number in Appendix A; IS: identification standard; RI: retention indices; compounds detected using both methods are written in bold. The bold line means overlap component with Table 3. *p*-value < 0.05. Data in the same column marked with different letters indicate significant differences at the 0.05 level. (Duncan’s multiple range test).

## Data Availability

The data presented in this study are available on request from the corresponding author.

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
