# Peer review of "Chemical Components of Dufour’s and Venom Glands in Camponotus japonicus (Hymenoptera, Formicidae)"

_insects, 2023, doi:10.3390/insects14070664_

Round 1
Reviewer 1 Report (Previous Reviewer 3)
In this new submission, the authors present a notably highly improved version compared to the previous ms on the subject. The English has been deeply revised being now much easier to read and understand even by non-specialists. Moreover, the number of flaws and errors is largely lower than those found in the previous ms. Scientifically, the work is sound and novel and the identification issue, one of my main concerns raised in the previous ms, has now been successfully assessed. Therefore, this new paper is acceptable for publication in Insects provided all edits are corrected in the review.
The English of this new ms has been deeply revised being now much easier to read and understand, even by non-specialists, than the previous submission.
Author Response
Great thanks for reviewer 1's approval!
Reviewer 2 Report (New Reviewer)
The study by Xu et al. on the chemical components of Dufour's and venom glands in Camponotus japonicus (Hymenoptera, Formicidae) is intriguing, well-written, and easy to follow. The authors have conducted a good number of replications, resulting in robust findings. The results and discussions are well presented, providing valuable insights. Overall, it is a knowledge incremental study.
Here are some specific suggestions for improvement:
- Line 88: It would be helpful if the geographical locations (GPS coordinates) where the ants were collected were provided.
- Line 106: The phrase "...chromatographic pure" should be deleted, and instead, the authors should mention the grade and purity of the hexane used.
- Line 114 to line 117: These lines should be deleted and merged with line 96 for better organization and flow of information.
- Line 313-314: The conclusion provided in these lines appears to be mere speculation. It would be better to revise it and instead provide a recommendation for future studies focusing on the role of undecane.
- Line 336: "n-undecane."
- Consider citing this work ‘Re-Analysis of Abdominal Gland Volatilome Secretions of the African Weaver Ant, Oecophylla longinoda (Hymenoptera: Formicidae) by Mekonnen et al., 2021
Author Response
Please see the attachment.

This manuscript is a resubmission of an earlier submission. The following is a list of the peer review reports and author responses from that submission.
Round 1
Reviewer 1 Report
This manuscript claims to describe the chemical contents of the Dufours and venom glands of an ant species, Camponotus japonicus. The authors dissected the glands and prepared hexane extracts, or collected volatiles from the dissected glands by solid phase microextraction, and then analyzed the extracts by GCMS. However, it is clear from their tables of results that they do not reallly know what they are doing in terms of how to identify compounds properly. That is, it appears as though they have completely relied on the instrument’s matches of their spectra with database spectra, many of which are clearly incorrect based on the mismatch between the proposed structures and the retention index values taken on a nonpolar HP5 column. The authors also state, L 395, “The compounds were identified according to the standard substances…” but this is clearly not true for many, possibly most of the compounds because so many of them have been misidentified, or identified on the basis of incomplete information (e.g., claiming to have identified the position and stereochemistry of double bonds in alkenes, acids, and esters, when this can only be done reliably by derivatization). Specific examples are as follows:
1. In table 1, the claim to have identified 8-heptadecene, but there is no indication at all how they identified the position of the double bond. The stereochemistry is also not stated. It is easily possible to determine both the position and stereochemistry by e.g., dimethyldisulfide derivatization.
2. First compound in Table 2 is cis-13-octadecenoic acid, with a claimed Kovats retention index (KI) of 3076, which is impossible on a nonpolar column. The position of the double bond also cannot be determined reliably from the mass spectrum only, and the double bond stereochemistry is not given. The same is true for the next compound, cis-vaccenic acid, where the KI is impossible for an acid of that chain length, and the double bond position cannot be determined reliably from the mass spectrum alone. Same problem with the third acid.
3. Table 2 lists the KI value for hexadecanol as 2127 (which is incorrect), but the KI value for the longer chain heptadecanol as 1905, i.e., the authors do not seem to have realized that it is clearly impossible for a shorter chain homolog to have a KI value GREATER than that of the longer chain homolog.
4. As mentioned, Table 2 lists the KI value for hexadecanol as 2127, but in Table 3, the value is given as 1958, and in Table 4, the KI value for the same compound is listed as 1891. Again, the authors do not seem to understand that the KI value for a compound is a physical constant, which has to be the same when the compound is run on the same column under standard conditions.
5. Tables 2 and 4 list 2(octadecyloxy)-ethanol, which is unlikely to be a natural product, and its molecular formula is given incorrectly as C31H64, and its KI value of 3321 is impossible.
6. Table 2, the KI value for 2-methyloctadecane is given as 2709, which is impossible. KI for 7-methylpentadecane is given as 1916, also clearly impossible. Further down, the authors state that diisoctylphthallate was present, without recognizing that this is not a natural product, but a well known plasticizer, i.e., a contaminant.
7. And so on…
8. Other examples from Table 4: KI value for 8-hexylpentadecane (which is unlikely to be a natural product based on biosynthetic considerations) is listed as 3437, which is clearly impossible for a 21 carbon hydrocarbon. Further on, the authors claim to have identified 4-pentafluoroethyl imidazole, which is clearly impossible as a natural product, i.e., no natural products from any organism, anywhere, contain a pentafluoroethyl group.
9. The authors have also used an incorrect formula for calculating KI values. The formula that they used is for isothermal runs. With temperature programming, the relationship simplifies to a linear one.
In short, many or most of the claimed identifications of compounds in this manuscript cannot be trusted because even at a superficial reading, so many of them are clearly incorrect or impossible, or both. Because the identifications and comparisons were the main topic of this manuscript, I have to recommend rejection.
Reviewer 2 Report
The experimental design is reasonable, the data analysis is correct, the result description is accurate and the discussion is sufficient.
Reviewer 3 Report
This ms deals with the identification of chemical compounds in the Dufour’s and venom glands of ants Camponotus japonicus. Although the chemical composition of the Dufour’s gland in many Camponotus spp. has been reported, so far there has been no comparative report of the chemical composition of both glands. Therefore, the work is novel in this regard and although this represents a sound value, the ms contains a number of flaws and errors, grammatical and non-grammatical, which markedly hampers the value of the ms. My main concern deals with the identification issue and how it was conducted. In MM there is no description on how the SPME method was done (type of fiber used, time of absorption, …), and this is an important subject for colleagues who want to repeat the work for similar or other purposes. Also, the authors should have commented in tables how they identified each compound, either by coinjection with a standard, by their retention index (which incidentally should not contain decimals), mass spectra in comparison to databases, or by combination thereof. I also wonder how the authors dealt with the quantification of high volatile chemicals, such as formic acid (compound 1, Table 3 and Fig. S3) and oxoacetic (commonly known as glyoxylic) acid (comp. 2, Table 3 and Fig. S3). These compounds were only apparent in the SPME method but they were poorly resolved and completely absent by the Hexane solution method since both chromatographic peaks overlaps with that of the solvent (hexane in the Hexane method). Why did the authors not tested a more volatile solvent, such as pentane in what could have been Pentane method, to see whether both chemicals appeared more readily separated and easier to integrate? Incidentally, in the queen caste (Fig. S3) 3-penten-2-one (comp. 3) peak is apparently much higher than acid 2, but in Table 3 the relative percentage of the acid is sensibly higher (11.27%) than that of the ketone (4.86).
English expressions and gramma should be largely revised by an English native expert or professionally proofread. F.i. and as examples, in L18 it is said ‘…are dominant components and their proportion shows obviously different among castes’, in L388 ‘the condition of mass spectrum was operated…’, or in L393 ‘…and formic acid were analyze with the sample instrument and procedure’, but there are many other along the ms.
Naming of compounds. Compound names (Tables 2,4) should follow IUPAC rules and the stereochemistry of the unsaturated chemicals should be cited as Z/E instead of cis/trans. Thus, cis-vaccenic acid should be named (Z)-11-octadecenoic acid, palmitoleic acid as (Z)-9-hexadecenoic acid, and primary alcohols should not be followed by ‘-1’, etc. In text, the first letter of the chemical names should not be capital. Homogenize naming the chemicals whenever possible, f.i. isopropyl palmitate should be preferentially named hexadecenoic acid, isopropyl ester. Very common compounds such as squalene and cholesterol, may be removed from Tables 2 and 4 since it is very unlikely they are active components of the gland secretions. Also, diisooctyl phthalate and 4-pentafluoroethyl imidazole are probably contamination products and should be removed and not included in statistics.
Table 2. The amount of each compound (ng) refers to the contents of the product per insect?. As cited above, retention indices should be given as whole number without
column as ‘Compound number’ so that each compound could be associated to the corresponding number in Supplementary files.
L63 It reads ‘we investigated the chemical composition of Dufour’s and venom glands of C. japonicus by means of chromatography and mass spectrometry (GC-MS)’. It is more precise to say ‘we investigated the chemical composition of Dufour’s and venom glands of C. japonicus by coupled gas chromatography-mass spectrometry (GC-MS)’.
L82. Cite tables in numerical order (Table 6 is cited after Table 2 and then Table 5…).
L87-88 The sentence ‘…n-Undecane was the dominant compound and the content is significant higher in gyne than that in other castes (Table 5)’ does not correspond to what appears in Table 5 wherein no significantly different values is apparent along the n-Undecane row. Please, correct.
L152-54 Comments of this paragraph of Table 1 refer to the ‘hexane solution method’ and the ‘secretions weight of the Dufour’s gland in gyne (11532.70 ng) was far higher than other castes’. However, only a list of ‘chemical components of Dufour’s gland secretions detected by SPME method’ is shown, and, moreover, no weights of the gland in the different castes appear at the table. The authors should be more careful about the title of the tables and their real contents
L160 Caption of Fig. 1. Insert ‘compounds’ between ‘secretion’ and ‘overlap’ to read ‘UpSet plots showing Dufour’s gland secretion compounds overlap’.
L167 To me there is no apparent faithful correspondence between Table 3 and Fig. S3, particularly in the relative percentage of some chemical components detected by SPME in castes major worker, gyne and queen. F.i. in the table the presence of comp. 5 in major worker amounts to 11.37% whereas comp. 10 is only 0.7% (?) but these values do not correspond to the apparent intensity of the corresponding peaks in the figure. The same happens with compounds 9 and 7 and others of the cited castes. Do the chromatograms of Fig. S3 belong to only one insect whereas Table 3 reflects the average amount of each compound 1-10 in all insects considered per caste?
Figures S1, S2, S3 and S4 do not contain any caption. Contents of the Supplementary Material should be specifically cited at the end of the text.
In Tables 5,6 data with an asterisk should be considered ‘significant’ or ‘highly significant’ rather than ‘remarkable’ or ‘extremely remarkable’.
In summary, to me this ms is not acceptable for publication in Insects in its present form. However, if the authors want to prepare a new revised ms taking into account all points and concerns raised above, the new ms could eventually be re-considered for publication. In my opinion, the new ms should be considered as a new submission.